# *Lilium philadelphicum* Flower as a Novel Source of Antimicrobial Agents: A Study of Bioactivity, Phytochemical Analysis, and Partial Identification of Antimicrobial Metabolites

**Shefali Singh [1], Vineeta Singh [1], Alaa Alhazmi [2,3], Bhartendu Nath Mishra [1,\*], Shafiul Haque [4,5],
R. Z. Sayyed [6,\*] and Kumari Sunita [7]**

[1] Department of Biotechnology, Institute of Engineering and Technology,
Dr. A.P.J. Abdul Kalam Technical University, Lucknow 226021, India; shefali220692@gmail.com (S.S.);
vscdri@gmail.com (V.S.)

[2] Medical Laboratory Technology Department, Jazan University, Jazan 45142, Saudi Arabia;
alaa.alhazmi@gmail.com

[3] SMIRES for Consultation in Specialized Medical Laboratories, Jazan University, Jazan 45142, Saudi Arabia

[4] Research and Scientific Studies Unit, College of Nursing and Allied Health Sciences, Jazan University,
Jazan 45142, Saudi Arabia; shafiul.haque@hotmail.com

[5] Faculty of Medicine, Görükle Campus, Bursa Uludağ University, Nilüfer 16059, Turkey

[6] Department of Microbiology, PSGVP Mandal's Arts, Science, and Commerce College, Shahada 425409, India

[7] Department of Botany, DDU Gorakhpur University, Gorakhpur 273009, India; ksunita78@gmail.com

\* Correspondence: profbnmishra@gmail.com (B.N.M.); sayyedrz@gmail.com (R.Z.S.)

**Abstract:** The members of the Liliaceae family are considered an excellent source of biologically active compounds. However, work on antimicrobial potential and characterization of the bioactive fractions of the *Lilium philadelphicum* flower is limited and needs to be explored. The present study reports the antimicrobial potential of the bioactive fraction extracted from the flower of *L. philadelphicum* (red lily) and partial characterization of the bioactive compound(s). The antimicrobial activity was tested against nine different Gram-positive and Gram-negative bacterial strains. The minimum inhibitory concentration (MIC) values of methanolic extract of the *L. philadelphicum* flower against *Acinetobacter bouvetii*, *Achromobacter xylosoxidans*, *Bacillus subtilis* MTCC 121, *Candida albicans* MTCC 183, *Klebsiella pneumoniae* MTCC 3384, and *Salmonella typhi* MTCC 537 were 25, 50, 12.5, 50, 100, and 50 µg/mL, respectively. The phytochemical analysis of the extract revealed the presence of phenols, flavonoids, tannins, terpenoids, glycosides, coumarins, and quinones. The cytotoxicity of the partially purified compound against the HepG2 cell line using MTT assay demonstrated up to 90% cell viability with a bioactive compound concentration of 50 µg/mL. However, the increase in the bioactive compound's concentration up to 1000 µg/mL resulted in nearly 80% cell viability. This minor decline in cell viability suggests the importance and suitability of the bioactive compound for therapeutic applications. Spectroscopic studies of the bioactive compound by UV-visible spectroscopy, FT-infrared spectroscopy, gas chromatography-mass spectrometry (GC-MS), as well as phytochemical analysis, suggested the presence of a terpenoid moiety, which may be responsible for the antimicrobial property of the *L. philadelphicum* flower.

**Keywords:** antimicrobial activity; characterization; GC-MS analysis; FTIR analysis; red lily; secondary metabolites

## 1. Introduction

Plants with the most diversity in volatile compounds comprise a substantial portion of biogenic hydrocarbons in the environment [1]. These volatile compounds, such as terpenes, phenolics, benzenoids, and fatty acid derivatives, along with nitrogen and sulfur-containing compounds produced as secondary metabolites, possess antimicrobial

properties and suggest that plants have enormous potential for providing novel therapeutic agents [2–4]. India, because of its geographical diversity, has a multitude of plants with various medicinal values. The antioxidant activities of plant-derived metabolites, along with the radical scavenging and redox potential of the phytochemicals, such as polyphenols, etc., are generally responsible for their therapeutic effect on human health [5]. However, because researchers are more focused on studies of the biological activities of higher plants, relatively few studies are reported that deal with the isolation, purification, and characterization of biologically active compounds from flowers.

The lily family, Liliaceae, consists of fifteen genera and about 705 known species of flowering plants within the order Liliales. The members of the Liliaceae family have been found to contain phytochemicals such as alkaloids, steroidal saponins, vitamins, and fatty acids, which are responsible for their biological activity [6]. *Lilium philadelphicum*, also known as the wood lily, Philadelphia lily, prairie lily, or western red lily, is a perennial species of lily native to North America. Flowers of various species of lily have been reported to possess broad-spectrum antimicrobial activity [6,7]. The polysaccharides isolated from *L. brownii* have been reported to significantly inhibit the growth of the melanoma B16 cancer cell line and lung cancer in mice [7]. Mitogenic and antifungal activities have also been reported from the compounds isolated from the *L. brownii* plant [8]. *L. longiforum* has been studied and used as an anti-inflammatory agent for the treatment of bronchitis and blood clotting during surgical procedures [8,9]. Lipid peroxidation and cholesterol oxidase enzyme inhibitor assays have been used to determine the bioactive compounds in the *L. longiforum* flower, which in turn shed light on its anecdotal medicinal use [9]. In this study, the antimicrobial potential of the bioactive fractions extracted from the flower of *L. philadelphicum* was tested against nine microbial strains, it was followed by phytochemical, spectroscopic, and cytotoxicity studies. The evaluation of the antimicrobial properties of the extracted volatile component(s) from the *L. philadelphicum* flower is expected to provide new dimensions in the treatment of infectious diseases.

## 2. Materials and Methods

### 2.1. Source of L. philadelphicum Flowers and Microbial Cultures

The *L. philadelphicum* flowers used in this study were collected from the campus of the Institute of Engineering and Technology, Dr. A.P.J. Abdul Kalam Technical University, Lucknow, Uttar Pradesh, India. Microbial cultures, namely, *Achromobacter xylosoxidans* and *Acinetobacter bouvetii*, were obtained from the culture repository of the institute. Other cultures such as *Bacillus pumilus* MTCC 1607, *Bacillus subtilis* MTCC 121, *Candida albicans* MTCC183, *Escherichia coli* MTCC 1304, *Klebsiella pneumoniae* MTCC 3384, *Salmonella typhi* MTCC 537, *Staphylococcus aureus* MTCC 96, and *C. albicans* were procured from the Microbial Type Culture Collection (MTCC), Chandigarh (Punjab), India.

### 2.2. Purification and Characterization of Bioactive Compounds from the Red Lily Flower

*L. philadelphicum* flowers collected from the college campus (Institute of Engineering and Technology, Lucknow, Uttar Pradesh, India) were used in this study. Flowers were dried under shade at 30–40 °C (Supplementary Information: Figure S1a) and subsequently powdered. The crude powder was extracted (Supplementary Information: Figure S1b) with various solvents viz. hexane, chloroform, methanol, and water in a ratio of 1:10 (crude powder:solvent ratio) at 28 °C. The extracts were filtered with Whatman filter paper, concentrated under vacuum, and stored at 4 °C until further use. The extraction yields of different solvents were determined using the following formula:

$$\text{Extract yield} = \frac{W1}{W2} \times 100 \tag{1}$$

where,

W1 is the net weight of the flower powder in gm after extraction, and
W2 is the total weight of the flower powder in gm taken for extraction.

### 2.3. Evaluating Antimicrobial Activity of the Red Lily Flower Extract

The antimicrobial activity of the crude methanolic extract of the lily flower was determined using the agar well diffusion method against *A. xylosoxidans*, *A. bouvetii*, *B. pumilus* MTCC 1607, *B. subtilis* MTCC 121, *C. albicans* MTCC183, *E. coli* MTCC 1304, *K. pneumoniae* MTCC 3384, *S. typhi* MTCC 537, and *S. aureus* MTCC 96 (Supplementary Information: Table S1; Figure S2). The 24 h old bacterial and fungal test cultures were grown on nutrient agar (NA) (Hi-Media, Mumbai, India) and potato dextrose agar (PDA) (Hi-Media, Mumbai, India), respectively. The wells of 9 mm diameter were bored and filled with 200 μL of the solvent extracts of the red lily flower, and the culture plates were incubated at 37 °C for 24 h for bacteria and 72 h for fungus. After the incubation, the plates were observed for the inhibition of the growth of the test organisms, and the diameter of the zone of inhibition from each plate was measured [10].

Phytochemical Screening

The methanolic extract of the *L. philadelphicum* flower was screened to examine the presence of chemical groups and active compounds, such as carbohydrates, saponins, phenols and tannins, coumarins, flavonoids, amino acids, glycosides, terpenoids, and quinones (Supplementary Information: Figure S3).

To detect the presence of carbohydrates, the flower extract was dissolved in 5 mL distilled water and filtered. The filtrate was hydrolyzed with dilute HCl and further neutralized with alkali and subsequently heated with Fehling's solution A and B and observed for the formation of a red precipitate of reducing sugars [11]. The examination of saponin was performed by the foam test. The flower extract (0.5 g) was vigorously mixed with 2 mL of water and observed for foam formation for more than 10 min as an indication of the presence of saponin [11]. The presence of phenols and tannins was detected by performing the ferric chloride test. Ferric chloride (0.5%) solution was added drop by drop to 2 mL of flower extract and observed for the formation of a bluish-black precipitate of phenols and tannins [11]. About 0.5 g of the moistened flower extract was placed into a test tube. The mouth of the test tube was covered with filter paper treated with 1 N NaOH solution. The treated test tube was placed in boiling water for a few minutes and examined for the formation of yellow color as an indication of the presence of coumarins [12]. To test the presence of flavonoids in the flower extract, a 10% lead acetate solution was added to the extract. The formation of yellow precipitate confirmed the presence of flavonoids [11]. The presence of amino acids in *L. philadelphicum* flowers was checked by employing a ninhydrin test. A few drops of ninhydrin solution were added to the flower extract, and the appearance of blue color indicated the presence of amino acid [13]. For the identification of glycosides, 1 mL of glacial acetic acid, a few drops of ferric chloride solution, and concentrated $H_2SO_4$ (mixed slowly through the sides of the test tube) were added to the flower extract and observed for the appearance of a reddish-brown ring of de-oxy sugars at the junction of the liquids [11]. For terpenoids identification in the flower extract, 2 mL of chloroform was added to 5 mL of the flower extract, and thereafter 3 mL of concentrated $H_2SO_4$ was added slowly and observed for the appearance of the reddish-brown color of terpenoids [11]. The flower extract was treated with a few drops of concentrated $H_2SO_4$ and observed for the formation of yellow color as an indication of the presence of quinones compound(s) [11]. The partial purification of the crude extract of the *L. philadelphicum* flower was performed by column chromatography using silica gel (mesh size 230–400) as a matrix. The column was eluted successively with an increasing gradient of methanol and chloroform. Further, the fractions were collected and examined by thin-layer chromatography (TLC) using a silica plate (TLC silica gel 60 F254) with methanol: chloroform (0.2% to 5%) as the mobile phase. Afterward, the TLC plate was removed from the solvent chamber, dried, and observed in the iodine chamber. The fractions having the same retention factor ($R_f$) values were pooled together and subjected to the bioassay [11].

## 2.4. Spectroscopic Analyses of Active Fraction

The chemical characterization of the active fraction containing volatile compounds was carried out with the help of UV-visible spectroscopy (PerkinElmer UV WinLab 5.2.0.0646/Lambda 25 spectrophotometer) and Fourier transform infrared spectroscopy IR (FT-IR PerkinElmer Model RX-1 spectrometer). Gas chromatography-mass spectroscopy (TSQ Quantum XLS) was performed to analyze the volatile compounds present in the active fraction.

## 2.5. Estimating Minimum Inhibitory Concentration (MIC) of the Partially Purified Compound

The MIC values of the partially purified compound were checked against Gram-positive and Gram-negative bacterial strains and fungus with erythromycin (E15) as a standard antibacterial drug according to standard protocol [14]. A 24 h old culture of each bacterial test strain ($5 \times 10^{-5}$ cells/mL) was grown in on NB; 200 µg/mL of the compound was taken as an initial concentration in the first test tube and was serially diluted. NB with and without erythromycin served as a negative and positive control, respectively. Another control, i.e., pure solvent (DMSO) only, was also included to observe the effect of the solvent on microbial growth. All the tubes were incubated at 28 °C overnight, and thereafter bacterial growth was observed [14].

## 2.6. Analyzing Cytotoxicity Using MTT Assay

In vitro cytotoxicity of the active compound against the HepG2 liver cancer cell line was analyzed by MTT assay as described by Mosmann [15] with minor modifications. A total of 100 µL of HepG2 cells suspension in a 96-well microtitre plate was incubated overnight at 37 °C in a $CO_2$ incubator. After 24 h, the medium was replaced by 100 µL of the fresh medium and treated with varying concentrations of the flower extract (0–1000 µg/mL). The microtitre plate was re-incubated at 37 °C under 5% $CO_2$ in the air for 24 h. The wells without flower extract served as a positive control. Further, MTT at 0.5 mg/mL concentration was added to the cell culture, and the plate was further incubated at 37 °C for 4 h. After the incubation, the culture supernatant was removed, and the cell layer was dissolved in DMSO (200 µL) and analyzed in a microplate reader (BioTek Instruments Inc, Winooski, VT, USA) at a test wavelength of 550 nm and a reference wavelength of 660 nm.

## 3. Results and Discussion

### 3.1. Antimicrobial and Phytochemical Screening of Solvent Extracts

The chemical investigation of the reddish-orange bioactive fraction extracted from *L. philadelphicum* revealed the presence of polar compounds. The methanolic and aqueous extract had greater extraction yields of 23.12 and 23.54%, respectively. Among the four solvents used during the extraction process, the yields obtained from chloroform and hexane extracts were 2.25 and 10.29%, respectively. The flower extract exhibited antimicrobial activity against *S. typhi* MTCC 537, *A. bouvetii*, *B. subtilis* MTCC 121, *A. xylosoxidans*, *K. pnemoniae* MTCC 3384, and *C. albicans* MTCC 183. However, the methanolic extract was not effective against *S. aureus* MTCC 96, *B. pumilus* MTCC 1607, and *E. coli* MTCC 1304. The aqueous extract of the *L. philadelphicum* flower exhibited antimicrobial activity against *B. subtilis*, *B. pumilus*, *K. pneumoniae*, *A. bouvetti*, and *C. albicans*. However, chloroform and hexane extracts were found slightly effective only against *K. pneumoniae* MTCC 3384 and *A. xylosoxidans*.

In most of the developing countries, including India, various infectious diseases are still creating challenges in the healthcare sector. To counteract such challenges, a variety of antimicrobials have been discovered to fight against the pathogens responsible for various diseases. These antimicrobials can be obtained as secondary metabolites from microbes, animals, or plants or can be synthesized chemically [16,17]. Due to the occurrence of wide-spectrum and multiple drug resistance (MDR) in pathogens towards existing antibiotics and the unpleasant side effects of the currently used antibiotics and synthetic

drugs, investigation of other sources, such as medicinal plants, for their antimicrobial properties is gaining importance [18]. The phytochemical analysis of the *L. philadelphicum* flower revealed the presence of flavonoids, tannins, glycosides, phenols, coumarins, and terpenoids (Table 1).

**Table 1.** Phytochemical screening of various extracts of red lily flower.

| Compounds | Extract | | | |
|---|---|---|---|---|
| | **Aqueous** | **Chloroform** | **Methanol** | **Hexane** |
| Saponins | − | − | − | − |
| Phenols | + | + | + | − |
| Glycosides | + | + | + | + |
| Flavonoids | + | + | + | + |
| Carbohydrates | − | − | − | − |
| Proteins | − | − | − | − |
| Coumarins | + | + | + | − |
| Quinones | + | − | + | − |
| Tannins | + | − | + | − |
| Terpenoids | + | − | + | − |

'+' Present; '−' Absent.

Several bioactive constituents have been reported from plants such as *Azardirachta indica* [19], *Senna alata* [20], and *Terminalia bellerica* [21]. In their efforts to search for potential antimicrobial compounds, Dontha et al. [22] reported the isolation of active constituents of *Ixora Javanica* DC flower extract and its phytochemical characterization. Similarly, in our study, the flower of *L. philadelphicum* was investigated, and the reddish-orange bioactive compound was isolated from the methanolic extract. The flower extract was found to be active against some Gram-negative/-positive bacteria and *Candida albicans.* Soliman et al. [17] reported the antifungal activity of a range of plants and suggested that they can be used as anti-candida agents after more in vivo investigations and suggested studies regarding the use of the nanostructured lipid system [23]. Following the encouraging results of the antimicrobial activities of the lily flower, we proceeded to the phytochemical analysis, purification, and identification of the bioactive compound.

### 3.2. Purification and Chemical Characterization of Bioactive Compound

Phytochemical analysis revealed that the *L. philadelphicum* flower is rich in phenols, flavonoids, quinones, tannins, glycosides, coumarins, and terpenoids. Spectroscopic and phytochemical analyses of the bioactive fraction from the lily flower suggest the presence of terpenoids. The flower's crude methanolic extract purified on silica column showed the presence of forty-eight fractions; these fractions were separately collected, their Rf values were determined, and all the fractions were subjected to antimicrobial bioassay. The initial fractions obtained from the silica column were more active against all the test organisms; however, fractions eluted later were active against one or two microbes only (Figure 1). The most active fraction was selected and subjected to partial characterization by UV-visible, FTIR, and GC-MS spectrum. The active fraction absorbed in the range of 290 to 360 nm and the pattern of the absorption suggested the presence of a double bond in the conjugation, which in turn reduces the possibility of the presence of saturated compounds (Figure 2).

The presence of strong absorption bands in the ultraviolet (UV)/visual (VIS) absorption spectrum is possibly due to $p{\rightarrow}p$ * or $n{\rightarrow}p$ * transitions [24]. In the performed experiments, the pattern of absorbance observed during the analysis suggests the presence of conjugation in the structure [24]. The IR spectrum explains different types of bonds, i.e., single, double, or triple among carbon, hydrogen, nitrogen, and oxygen atoms, which have diverse vibrational frequencies [25].

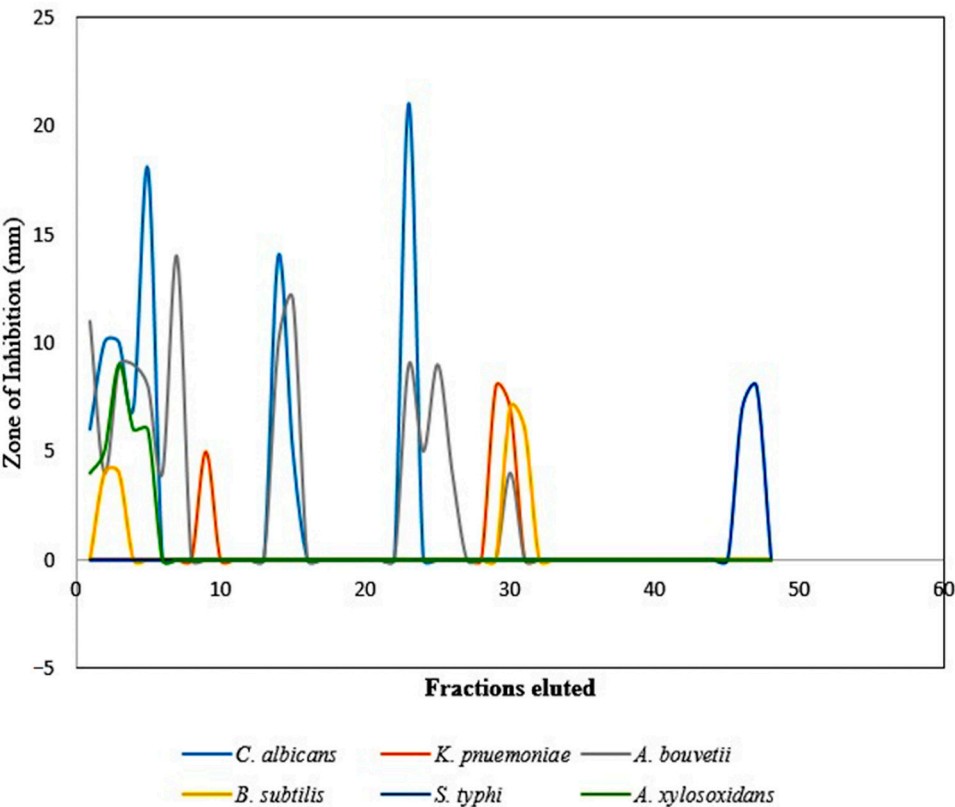

**Figure 1.** The activity profile of the column passed fractions showing biological activity, which were pooled and used for further purification processes.

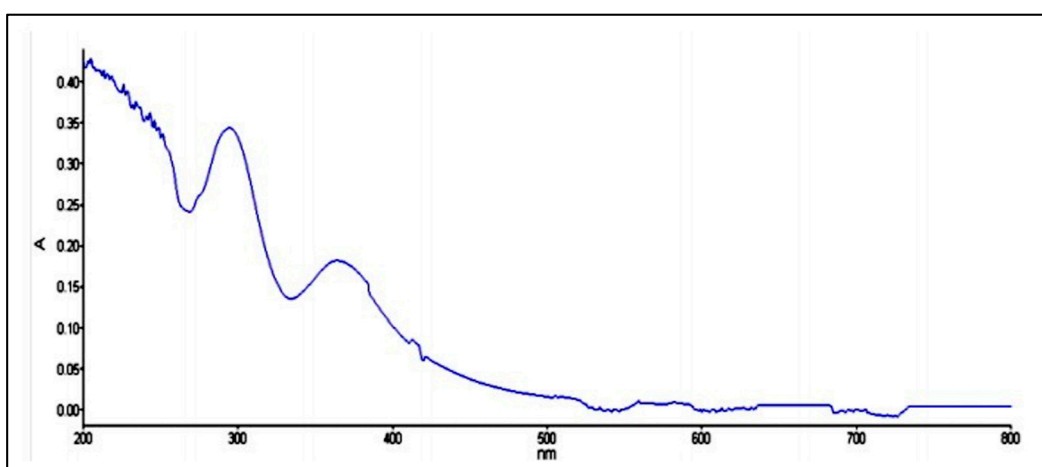

**Figure 2.** UV profile of the active fraction dissolved in methanol showing strong absorption in the ultraviolet range.

The FTIR spectrum of the samples exhibits bands at 2994.75 and 2911.83 cm$^{-1}$, which confirms the CH stretching (Figure 3). The band at 1435.83 cm$^{-1}$ indicates the aromatic or heteroatomic C-C stretching vibrations. The absorption band at 1309.05 cm$^{-1}$ corresponds to the C=O moiety in the compound. The frequencies were identified at 951.52, 696.04, and 666.66 cm$^{-1}$ corresponds to =C-H bending vibrations.

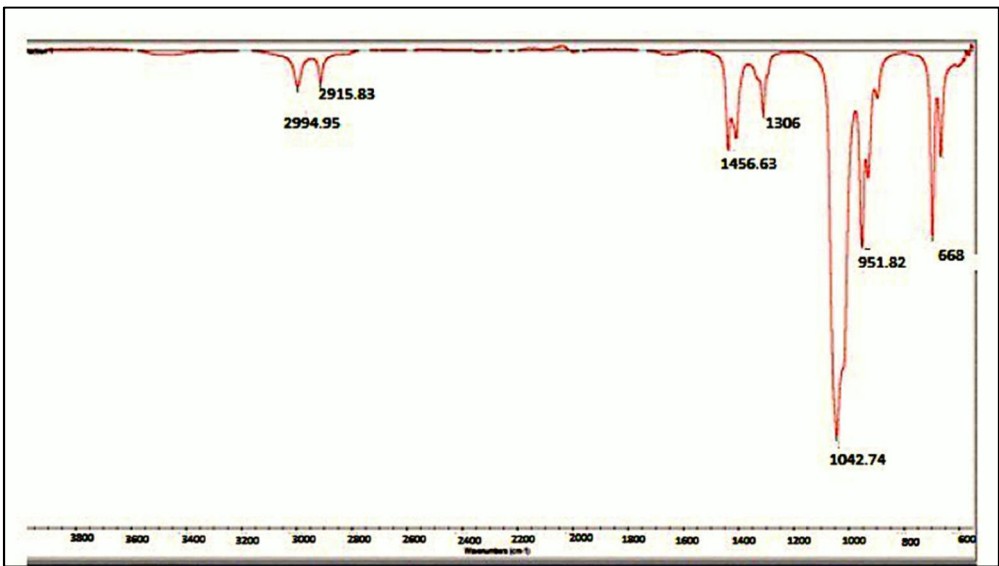

**Figure 3.** Sharp peaks in the FTIR spectrum of the active compound in methanol suggesting the presence of specific functional moieties in the active compound.

The GC-MS spectrum shows nine major ideal peaks with different retention times as 13.30, 18.93, 24.97, 28.04, 30.91, 33.46, 36.02, 40.77, and 44.62 min (Figure 4).

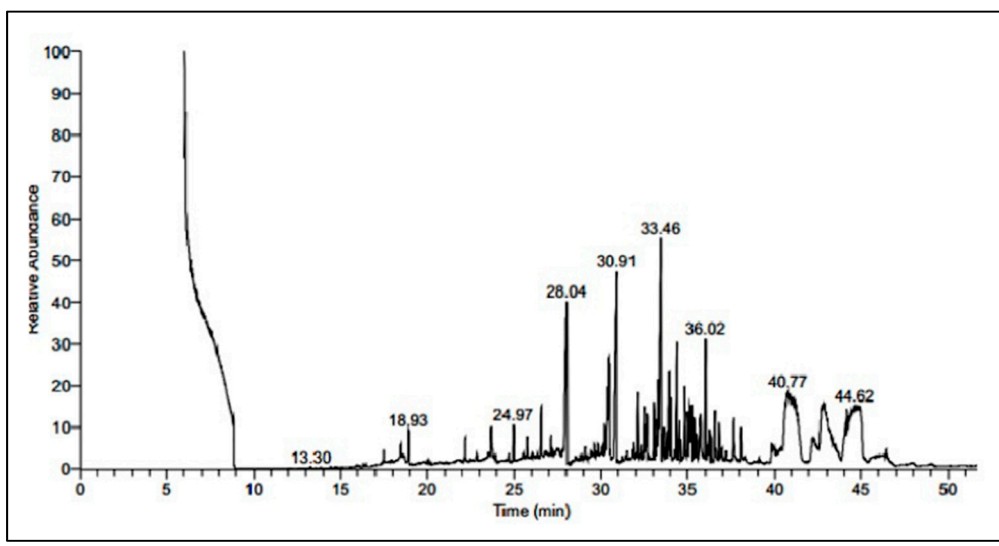

**Figure 4.** GC-MS profile of the active fraction showing peaks at different retention times with variable abundance reflecting the presence of different compounds in the active fraction.

Further, GC-MS of the sample was recorded on TSQ Quantum XLS for the identification of the bioactive component(s) of the flower. The generated spectrum was processed for major ideal peaks found during the analysis and compared with the mass spectrum from the library standards available in the database (NIST library). The GC-MS spectrum shows nine peaks at different retention times (RT) with different probability factors (PF), which in turn reflects the presence of nine different compounds. These compounds are chloculol (RT 13.30 min; PF 68.43; $m/z$ 205.1); (+ −)-5-Hydroxy-6-(1-hydroxy ethyl)-2, 7-dimethooxynapthoquinone (RT 18.93 min; PF 2.93; $m/z$ 263.2); 1,2-Benzenedicarboxylic acid, bis(2-methylpropyl) ester (RT 24.93 min; PF 30.42; $m/z$ 149.0); 12-methoxy-2-trimethylsilyloxy-19-nor-5ápodocarpa-1,3,8,11,13-pentaene (RT 28.04 min; PF 50.12; $m/z$ 314.1); 3-Acetoxy-7á-(acetylthio)-17á pregna-3,5-diene-21,17-carolactone (RT 30.91 min; PF 26.29; $m/z$ 341.3); 2-(Dimethyl phenylsiyl) hepta-1,5,diene-4-one (RT

33.46 min; PF 19.82; *m/z* 135.1); 3-[(t-butyl diphenylsilyl) oxy]-4-benzyloxy-6-acetoxy-6-vinyl-1-bromo-1-cyclohexene (RT 36.02 min; PF 43.88; *m/z* 399.4); 1, 4, 5, 7, tetrahrdoxy-2-methylanthaquinone (RT 40.77 min; PF 46.34; *m/z* 207.2), and silane (RT 44.62 min; PF 56.42; *m/z* 129.0) [20].

### 3.3. Estimating Minimum Inhibitory Concentration (MIC) and Cytotoxicity of the Active Fraction

The MIC of the partially purified compound against *Candida albicans* MTCC 183, *Achromobacter xylosoxidans*, and *Salmonella typhi* MTCC 537 was 50 µg/mL. The purified compound was found to be less active (shows MIC 100 µg/mL) against *Klebsiella pneumoniae* MTCC 3384. However, it was found to be more effective against *Acinetobacter bouvetii* and *Bacillus subtilis* MTCC 121 with MIC values of 25 and 12.5 µg/mL, respectively.

Further, the cytotoxicity analysis of the bioactive compound on the HepG2 cancer cell line employing MTT assay demonstrated approximately 9.7% cell inhibition at 50 µg/mL concentration and approximately 20% cell inhibition at 1000 µg/mL concentration (Figure 5).

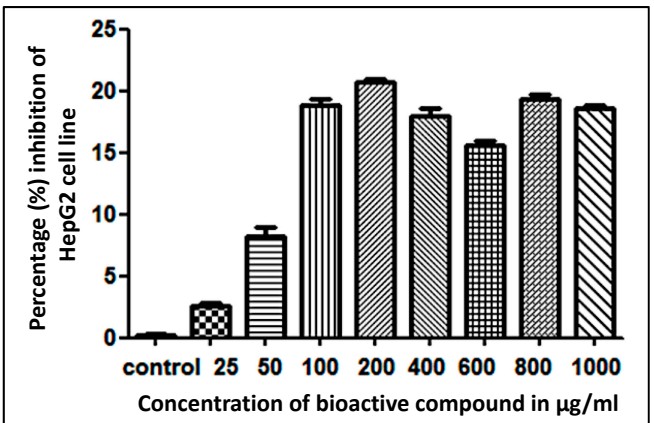

**Figure 5.** Cytotoxicity profile of the active fraction against human liver cancer HepG2 cell line.

The phytochemical analysis of the *L. philadelphicum* flower is in line with earlier findings, where it has been reported that carbonyl compounds such as terpenoids, especially monoterpenes [C10], sesquiterpenes [C15], alcohols, aldehydes, acyclic esters or lactones, and exceptionally nitrogen and sulfur-containing compounds, coumarins, and homologs of phenylpropanoids, exhibit a wide spectrum of biological activity [26]. The active fraction may contain terpenoids, as evidenced by the spectroscopic and phytochemical analyses. The terpenoids in the active fractions might be responsible for the antimicrobial properties of the *L. philadelphicum* flower. To the best of our knowledge, this is the first study deciphering the antimicrobial activity of the *L. philadelphicum* flower.

## 4. Conclusions

The potent and broad-spectrum antimicrobial properties of the *L. philadelphicum* (red lily) flower investigated in the present study suggest its therapeutic potential against Gram-positive and Gram-negative pathogens. The phytochemical analyses reveal that the polar organic solvents are more efficient in the extraction process and thus suggest the polar nature of the metabolites present in the flower. Further, the cytotoxicity assay concludes that approximately 90% of cells are viable up to 50 µg/mL concentration of the partially purified compound from the *L. philadelphicum* flower. In conclusion, the antimicrobial activity of the *L. philadelphicum* flower, along with cytotoxicity results, suggests the future application of this flower for antimicrobial purposes in the treatment of various infectious diseases. Further research in the direction of the complete characterization of the active compound and in vivo mechanistic studies in animal models may provide better insight into the understanding of the identification and development of suitable bioactive agents to treat various infectious diseases.

**Supplementary Materials:** The following are available online at https://www.mdpi.com/article/10.3390/su13158471/s1, Table S1. Antimicrobial activity of the extracts of *L. philadelphicum* tested against different microorganisms: Zone of Inhibition obtained using various solvent extracts. Figure S1. (a) Dried flowers of lily; (b) example of crude extract of lily flower. Figure S2. Some examples of Zone of Inhibition (ZOI) obtained from various solvents (methanol/hexane/aqueous/chloroform) extract of the flower. Figure S3. Phytochemical analysis of the methanolic extract of red lily flower.

**Author Contributions:** Conceptualization, V.S., B.N.M. and R.Z.S.; Formal analysis, V.S., S.S. and A.A.; Investigation, V.S., S.S., A.A. and S.H.; Methodology, V.S., S.S., A.A. and S.H.; Supervision, V.S., B.N.M. and R.Z.S.; Writing—review & editing, V.S., B.N.M., S.H., R.Z.S. and K.S. All authors have read and agreed to the published version of the manuscript.

**Funding:** No direct or indirect funding was available for this study.

**Institutional Review Board Statement:** Not applicable.

**Informed Consent Statement:** Not applicable.

**Data Availability Statement:** Not applicable.

**Acknowledgments:** The authors are grateful to the laboratory facility provided by the Department of Biotechnology, Institute of Engineering and Technology, A.P.J. Abdul Kalam Technical University, Lucknow (UP) India. The authors (A.A. and S.H.) sincerely acknowledge the support from Jazan University, Saudi Arabia, for providing access to the Saudi Digital Library.

**Conflicts of Interest:** The authors declare no conflict of interest.

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
