# Peer review of "Lilium philadelphicum Flower as a Novel Source of Antimicrobial Agents: A Study of Bioactivity, Phytochemical Analysis, and Partial Identification of Antimicrobial Metabolites"

_sustainability, doi:10.3390/su13158471_

Round 1
Reviewer 1 Report
The manuscript has some value in the context of searching new antimicrobial agent however analytical procedures are serious flaw of this study. Moreover some parts of manuscript are trivial and too general and have no scientific value. The examples have been included below:
- The introduction should be improved e.g. Line 39 – 49 – too general and have no scientific value, line 61 - exaggerated sentence
- Yield is incorrectly assessed because sample powder can be retained during filtration.
- Line 99-124: have no scientific value. General tests to assess various groups of compounds are very misleading and are not informative.
- Line 130: “ TLC was developed in the iodine chamber” – why iodine was used during TLC?
- Section 2.4. No component was identified. Authors only registered spectrum or profile of fraction.
- “The investigation on the reddish-orange bioactive compound extracted from philadelphicum reveals the presence of polar compounds.” – completely unclear sentence.
- Results and Discussion section need strong reedition. There are a lot of unnecessary information herein, e.g. Line 173-180: they should be a part of Introduction because they justified the aim of the study; line 186-189 and 195-197 are unnecessary because they are not linked with Results obtained by Authors.
- Line 206: “The active fraction absorbed in the range of 290 to 360 nm suggesting the presence of a double bond in the conjugation” – is not possible to draw such conclusion based on UV-VIS spectrum
- Line 212-213: is not possible to draw such conclusion based on UV-VIS spectrum
- 1: Flavonoids and glycosides were present in hexane extract? They are strongly polar.
- 1 is completely unclear
- 4. Much more components are visible on chromatogram. Moreover, GC allows to assess only volatile components.
Author Response
REVIEWER 1 Report
The manuscript has some value in the context of searching for new antimicrobial agents however analytical procedures are a serious flaw of this study. Moreover, some parts of the manuscript are trivial and too general and have no scientific value. The examples have been included below:
RESPONSE: We (all authors) thank the reviewer for his precious time for reading our manuscript and suggesting changes/corrections. This has improved the quality of the manuscript up to a great extent. The suggested changes/ corrections have been incorporated throughout the revised manuscript, and we hope that these corrections will meet the reviewer’s approval.
- The introduction should be improved e.g. Line 39 – 49 – too general and have no scientific value, line 61 - exaggerated sentence.
RESPONSE: As suggested by the esteemed reviewer, all the general lines (text) have been removed in the revised manuscript and some pertinent information has been incorporated in the revised manuscript.
- Yield is incorrectly assessed because sample powder can be retained during filtration.
RESPONSE: Yes, we appreciate the reviewer’s vigilant comment as he has raised a very valid point. However, we would like to state that during any filtration process involving powder sample, a fraction must be absorbed/retained on the filter paper, which is generally considered as method related handling error and it is applicable to all processes/batches involve and same for all, hence must be ignored; and, we practiced the same.
- Line 99-124: have no scientific value. General tests to assess various groups of compounds are very misleading and are not informative.
RESPONSE: We agree with the reviewer’s idea. However, the results of phytochemical testing provide a preliminary estimate of the active component present in the sample. Although, there are many methods of performing the phytochemical analysis; however, herein we have used the most commonly reported test to reduce the chances of misleading results.
- Line 130: “ TLC was developed in the iodine chamber” – why iodine was used during TLC?
RESPONSE: In order to observe the unsaturation present in the compounds generally iodine chamber or UV chamber are used. Herein, we used the iodine chamber.
- Section 2.4. No component was identified. Authors only registered spectrum or profile of fraction.
RESPONSE: We completely agree with the reviewer’s concern. However, we would like to state that as mentioned in the title of the manuscript, only preliminary or partial purification is reported in the current manuscript. Whereas, the purification of the compound(s) is on our priority list and will be done soon once will resume our work, as due to COVID, everything is on halt here in most part of India, and only limited or urgency related research work is under progress. We tried to expedite this purification work using commercial services, but all affected due to COVID; but, will complete and report the chemical characterization of the active compounds using UV, IR, Mass, and a range of NMR spectra in our future publication.
- “The investigation on the reddish-orange bioactive compound extracted from philadelphicum reveals the presence of polar compounds.” – a completely unclear sentence.
RESPONSE: As suggested by the learned reviewer, the suggested text has been rephrased in the revised manuscript.
- Results and Discussion section need strong reedition. There is a lot of unnecessary information herein, e.g. Line 173-180: they should be a part of Introduction because they justified the aim of the study; line 186-189 and 195-197 are unnecessary because they are not linked with Results obtained by Authors.
RESPONSE: As suggested by the esteemed reviewer, the Results and Discussion section has been modified in the light of the given comment and has been made more focused in the revised submission.
- Line 206: “The active fraction absorbed in the range of 290 to 360 nm suggesting the presence of a double bond in the conjugation” – is not possible to draw such conclusion based on UV-VIS spectrum.
RESPONSE: We appreciate the reviewer’s concern. However, the involvement of double bonds in the conjugation is predicted from the pattern of absorption, not on the basis of wavelength. The statement has been corrected in the revised version of the manuscript.
- Line 212-213: is not possible to draw such a conclusion based on the UV-VIS spectrum
RESPONSE: Earlier, the statement was unclear leading to infer misleading information, however, we have corrected the same in the revised submission. Now, we would like to humbly state that, from the UV-VIS spectrum profile we are predicting the presence of conjugation, which is also supported by previous reports. Further, predictions can be made generally by NMR analysis or chemical characterization. As per the suggestion, in order to avoid misleading information, the statement has been rephrased and modified in the revised version of the manuscript.
- Flavonoids and glycosides were present in hexane extract? They are strongly polar. is completely unclear
RESPONSE: Yes, in general, and as per our own observation and experience of the last 15 years, most of the flavonoids and glycosides are polar in nature. However, the presence of alkyl derivatives in these creates the tendency to get extracted in low-/non- polar solvents.
- Much more components are visible on chromatogram. Moreover, GC allows to assessment of only volatile components.
RESPONSE: As stated earlier and clarified in the title of the manuscript that we are reporting partial characterization; and we are not denying the possibilities of many more components in the chromatogram. Herein, we have only considered/discussed the components which were present in abundance in the chromatogram. Other components will be explored in future studies during further characterization. We agree GC allows only volatile components, which are the focus of our study.
Reviewer 2 Report
Dear authors, thank you very much for your research revealing the modern basis for ansient traditions.
- What is the main question addressed by the research?
The main question addressed is the search for new antimicrobial compounds based on the ancient experience of Ayurveda. This seems important taking into account the absence of new effective compounds in the pipeline.
- Is it relevant and interesting?
Yes
- How original is the approach?
The approach to the study of antimicrobial compounds cannot be fully original. The methods used could be improved or changed to alternatives. What is important - the fact that the experience collected for centuries is revised with modern approaches and it shows promise for future research and fight with the infections
- What does it add to existing publications on related topics?
There are many publications on the search for antimicrobial properties of plants and every new one is important and may lead to further studied by different researches and finally bring positive results to clinical medicine
- Is the paper well written?
Yes
- Are the conclusions consistent with the evidence and arguments presented?
Yes
Author Response
REVIEWER 2 REPORT
Dear authors, thank you very much for your research revealing the modern basis for ansient traditions.
- What is the main question addressed by the research?
The main question addressed is the search for new antimicrobial compounds based on the ancient experience of Ayurveda. This seems important taking into account the absence of new effective compounds in the pipeline.
- Is it relevant and interesting?
Yes
- How original is the approach?
The approach to the study of antimicrobial compounds cannot be fully original. The methods used could be improved or changed to alternatives. What is important - the fact that the experience collected for centuries is revised with modern approaches and it shows promise for future research and fight with the infections
- What does it add to existing publications on related topics?
There are many publications on the search for antimicrobial properties of plants and every new one is important and may lead to further studied by different researches and finally bring positive results to clinical medicine
- Is the paper well written?
Yes
- Are the conclusions consistent with the evidence and arguments presented?
Yes
Authors response: The authors are very much thankful to the Reviewer for appreciating the present research work
Reviewer 3 Report
- More information needed for all Figure legends. Currently, it is very poorly written
- The discussion chapter needs to be separated from the Results section and expand more to explore broader implications of the results
- The introduction chapter is too short and needs to add the recent references
- Statistics need to apply for data presented in figure 5
- For inhibition experiments, agar plate pictures need to be added and more data needed from measurements in stead of just positive and negative
Author Response
REVIEWER 3 REPORT
We (all authors) thank the reviewer for his precious time for reading our manuscript and suggesting changes/corrections. This has improved the quality of the manuscript up to a great extent. The suggested changes/ corrections have been incorporated throughout the revised manuscript, and we hope that these corrections will meet the reviewer’s approval.
- More information needed for all Figure legends. Currently, it is very poorly written.
RESPONSE: As suggested by the learned reviewer, updated detailed figure legends have been provided in the revised manuscript, as:
Figure 1. The activity profile of the column passed fractions showing biological activity, which were pooled and used for further purification process.
Figure 2. UV profile of the active fraction dissolved in methanol showing strong absorption in the ultra-violet range
Figure 3. Sharp peaks in the FTIR spectrum of the active compound in methanol suggesting the presence of specific functional moieties in the active compound
Figure 4. GC-MS profile of the active fraction showing peaks at different retention times with variable abundance reflecting the presence of different compounds in the active fraction
Figure 5. Cytotoxicity profile of the active fraction against human liver HepG2 cancer cell line.
- The discussion chapter needs to be separated from the Results section and expand more to explore broader implications of the results
RESPONSE: As per the journal guidelines, these sections to be together, and there is no objection from other reviewers for such changes. However, as required the Results and Discussion section has been modified in the revised submission.
- The introduction chapter is too short and needs to add the recent references
RESPONSE: We thank the learned reviewer for this comment, as suggested the introduction section has been modified and the information provided is supported with recent references in the revised manuscript.
- Statistics need to apply for data presented in Figure 5.
RESPONSE: We tried to follow your suggestion, but a majority of the published articles reporting the cytotoxicity data, wherein phytochemical testing has been done, are presented in this manner. So, herein we haven’t changed anything in Figure 5. For example, you can refer. Process Biochemistry, 2019, 87, 138-144. https://doi.org/10.1016/j.procbio.2019.08.024
- For inhibition experiments, agar plate pictures need to be added and more data needed from measurements instead of just positive and negative
RESPONSE: The asked data have been incorporated as a Supplementary Information file in the revised submission.
Round 2
Reviewer 1 Report
Biological assays have some value but phytochemical analysis are very poor and should be regarded as strong preliminary. It should be clearly reflected in title. Authors should precise The other remarks below:
- Line 135: “TLC was developed in the iodine chamber (…)”
Authors RESPONSE: In order to observe the unsaturation present in the compounds generally iodine chamber or UV chamber are used. Herein, we used the iodine chamber.
Comment: This explanation has no sense. TLC should be performed without access of iodine and after drying TLC plate should be treated with iodine. Moreover, in what purpose “unsaturation present in the compounds” was observed?
- Previous remark: Section 2.4. No component was identified. Authors only registered spectrum or profile of fraction.
Authors RESPONSE: “We completely agree with the reviewer’s concern. However, we would like to state that as mentioned in the title of the manuscript, only preliminary or partial purification is reported in the current manuscript. (…) We tried to expedite this purification work using commercial services, but all affected due to COVID; but, will complete and report the chemical characterization of the active compounds using UV, IR, Mass, and a range of NMR spectra in our future publication”.
Comment: I accept Authors’ explanation and it is good that the title of section has been changed on „Characterizing the bioactive fractions”. However, spectroscopic analysis of fraction give no essential information (the same remark for line 216- 239). It is obvious, that such complex mixtures such plant extract contain “double bond in the conjugation”, „conjugation in the structure” „the aromatic or heteroatomic C-C stretching vibrations”. Remove this part of investigation or move it to Supplementary.
- Previous remark: Line 173-180: they should be a part of Introduction because they justified the aim of the study; line 186-189 and 195-197 are unnecessary because they are not linked with Results obtained by Authors.
Comment: These parts of Discussion section have not been changed.
- Previous remark: Much more components are visible on chromatogram. Moreover, GC allows to assessment of only volatile components.
Authors RESPONSE: We agree GC allows only volatile components, which are the focus of our study.
Comment: No mention in the manuscript that the study was focused on volatile components. Moreover, such extraction isolate not only volatile compounds.
- Line 244: “GC-MS spectrum shows nine major ideal peaks with different retention times as 13.30, 18.93, 24.97, 28.04, 30.91, 33.46, 36.02, 40.77 and 44.62 min”
It is hardly agree that peak at 13.3 min. is „major” and peaks at 40.77 and 44.62 min are absolutely not ideal (their shape suggest that few components coeluted).
- Fig.1. What is shown on X-axis? Time at which the fraction was collected?
Author Response
Dear Sir
We (all authors) are grateful to the learned reviewer for reviewing our manuscript and for suggesting the revision. Further, we are thankful to the esteemed reviewer for his valuable comments, which were really valuable and have improved the quality of the manuscript for wider dissemination of the information provided in the article and broader readership.
As suggested, the corrections/ changes in the revised manuscript have been shown with track-change/RED color text. Please find the revised version (manuscript with changes highlighted and clean version) of the manuscript and point-by-point responses (in BLUE text) to the reviewer’s comments.
I also hope that now our REVISED manuscript is in an acceptable format.
Thank you,
Sincerely,
*Corresponding authors
Prof. Riyaz Sayyed & Prof. Bhartendu Nath Mishra
RESPONSE TO REVIEWER’S COMMENTS
Suggestions & Comments to the authors:
Biological assays have some value but the phytochemical analysis is very poor and should be regarded as strong preliminary. It should be clearly reflected in the title. The authors should precise The other remarks below:
- Line 135: “TLC was developed in the iodine chamber (…)”
Authors RESPONSE: In order to observe the unsaturation present in the compounds generally iodine chamber or UV chamber is used. Herein, we used the iodine chamber.
Comment: This explanation has no sense. TLC should be performed without access of iodine and after drying TLC plate should be treated with iodine. Moreover, for what purpose “unsaturation present in the compounds” was observed?
RESPONSE:
The text mentioned in the manuscript: “Further, the fractions were collected and examined by thin-layer chromatography (TLC) using a silica plate (TLC silica gel 60 F254) with methanol: chloroform (0.2% to 5%) as the mobile phase. TLC was developed in the iodine chamber and the fractions having the same retention factor (Rf) values were pooled together and subjected to the bioassay [10].”
The text provided in inverted “………” comma presents the methodology of TLC performed. Herein, readymade TLC silica gel 60 F254 plates were used for the spotting purpose; the crude and methanol: chloroform (0.2% to 5%) solvent system ran over the plate for separating the compounds based on their polarity. This was performed without using iodine. However, when the solvent system covered around 90% of the TLC, the plate was removed and dried. Further, to analyze the presence of number of components in the sample, the plate was examined under UV/florescence light to confirm the presence of unsaturated/florescent compounds. Likewise, TLC plates were subjected to iodine treatment in iodine chamber to confirm the presence of unsaturated compounds. Possibly the term “developed” used in the previous version of the manuscript created some mis-understanding. This term has been changed in the revised submission.
- Previous remark: Section 2.4. No component was identified. Authors only registered spectrum or profile of fraction.
Authors RESPONSE: “We completely agree with the reviewer’s concern. However, we would like to state that as mentioned in the title of the manuscript, only preliminary or partial purification is reported in the current manuscript. (…) We tried to expedite this purification work using commercial services, but all affected due to COVID; but, will complete and report the chemical characterization of the active compounds using UV, IR, Mass, and a range of NMR spectra in our future publication”.
Comment: I accept Authors’ explanation and it is good that the title of section has been changed on „Characterizing the bioactive fractions”. However, spectroscopic analysis of fraction give no essential information (the same remark for line 216- 239). It is obvious, that such complex mixtures such as plant extract contain “double bond in the conjugation”, „conjugation in the structure” „the aromatic or heteroatomic C-C stretching vibrations”. Remove this part of the investigation or move it to Supplementary.
RESPONSE: We thank the esteemed reviewer for this comment. As per reviewer’s suggestion, we have changed the title of the sub-section as “Spectroscopic analyses of active fractions”. Hope, this will give more precise information which we have covered in this manuscript.
Regarding suggestion of moving characterization data into Supplementary, we (all authors) were trying to figure out how we can shift some of the characterization data into Supplementary Information file, and we realized that, if will do it, nothing concrete will be remaining in the main text of the manuscript. Hence, to clarify the seemingly absurd text we have provided statements how we have down size or rationalize the current findings; and that is why we have written in the title that we are reporting the partial purification. And, the same information has been incorporated in the Results and Discussion section of the manuscript with some modifications.
“As established, most of the bioactive compounds from plant sources generally have unsaturation tendency, for example phenolic compounds. Hence, to confirm the presence of unsaturated compound(s) in the active fraction(s), the sample was subjected to iodine treatment. As we know that the double bonds present in the compound may be in conjugation or non-conjugation, which was further confirmed by UV spectrum analysis. Later, the active fraction was subjected to IR analysis to identify the prominent functional groups present in the bioactive compound. Whereas, LCMS/GCMS of the active fraction gave the idea about the molecular weight/property of the compound. However, we failed to perform the NMR studies for detailed structural properties, but it is on our priority list; will report our structural findings and further computational binding studies in our future publications.
- Previous remark: Line 173-180: they should be a part of Introduction because they justified the aim of the study; line 186-189 and 195-197 are unnecessary because they are not linked with Results obtained by Authors.
Comment: These parts of Discussion section have not been changed.
RESPONSE: The lines 194-197, “Although, the researchers are more focused on the studies on biological activities of higher plants, however, relatively scanty studies are reported dealing with the isolation, purification, and characterization of biologically active compounds from flowers.” have been shifted to the introduction section of the revised version of manuscript
- Previous remark: Much more components are visible on chromatogram. Moreover, GC allows to assessment of only volatile components.
Authors RESPONSE: We agree GC allows only volatile components, which are the focus of our study.
Comment: No mention in the manuscript that the study was focused on volatile components. Moreover, such extraction isolate not only volatile compounds.
RESPONSE: We noticed this discrepancy and needful text has been incorporated in the revised version of the manuscript.
- Line 244: “GC-MS spectrum shows nine major ideal peaks with different retention times as 13.30, 18.93, 24.97, 28.04, 30.91, 33.46, 36.02, 40.77 and 44.62 min”
It is hardly agree that peak at 13.3 min. is „major” and peaks at 40.77 and 44.62 min are absolutely not ideal (their shape suggest that few components coeluted).
RESPONSE: We are not claiming that 13.3 is the major peak, we discussed all the nine peaks and the possibility of the compounds on the basis of the database (NIST library) results.
- Fig. 1. What is shown on X-axis? Time at which the fraction was collected?
RESPONSE: Figure 1 shows the biological activity exhibited by the eluted fractions of column chromatography. x-axis depicts the number of fractions collected and the y-axis shows that the activity (zoi) by a particular fraction.
****

Reviewer 3 Report
Thank you authors for revising the manuscript and now it is improved than old version. However, I can see errors in Figure 5. I am not satisfied with the explanations of using statistics. The number of replicates and repeats need to mention in the figure legend as well as in the methods. The bars of the figure needs to explain in x-axis.
Author Response
RESPONSE TO REVIEWER’S COMMENTS
Comments and Suggestions for Authors
Thank you, authors, for revising the manuscript and now it is improved than the old version. However, I can see errors in Figure 5. I am not satisfied with the explanations of using statistics. The number of replicates and repeats need to mention in the figure legend as well as in the methods. The bars of the figure need to explain in the x-axis.
RESPONSE: We thank the learned reviewer for pointing this issue; needful correction has been done in Figure 5 (appended below), also statistical explanation has been provided for reference.
|
Table Analyzed |
Data 1 |
||
|
|
|||
|
One-way analysis of variance |
|||
|
P value |
< 0.0001 |
||
|
P value summary |
*** |
||
|
Are means signif. different? (P < 0.05) |
Yes |
||
|
Number of groups |
9 |
||
|
F |
347.9 |
||
|
R squared |
0.9936 |
||
|
|
|||
|
ANOVA Table |
SS |
df |
MS |
|
Treatment (between columns) |
1478 |
8 |
184.7 |
|
Residual (within columns) |
9.557 |
18 |
0.5309 |
|
Total |
1487 |
26 |
Figure 5. Cytotoxicity profile of the active fraction against human liver cancer HepG2 cell line.
